# WORLDCRAFT

## ABSTRACT

We present Worldcraft, a hybrid implicit method for generating vast, interactive 3D worlds at unprecedented scale and speed by modeling them as exchangeable sequences of latent 3D objects. In contrast to existing methods that produce limited scenes, Worldcraft's novel approach constructs expansive environments comprising thousands of elements, extending to over a million objects in seconds, on a single GPU. The resulting created *worlds* are defined in terms of possessing certain essential properties: Object Individuality, Collective Semantics, and Expandability. To achieve this with both speed and scale, we conceptualize world generation as a *set generation problem*, introducing three key technical innovations: (i) Hierarchical and Exchangeable Sequence Modeling ensures Object Individuality while capturing Collective Semantics; (ii) Hybrid Implicit Generation Method enables rapid creation of vast worlds, supporting both Scale and Expandability; and (iii) Multi-level Indexing Functions allow efficient manipulation across scales, reinforcing Collective Semantics and enabling on-demand generation for Speed and Expandability. We demonstrate Worldcraft's capabilities using Minecraft as a test-bed, generating complex, interactive environments that users can explore. However, this approach is applicable to any suitable platform, potentially revolutionizing various applications in 3D environment generation.

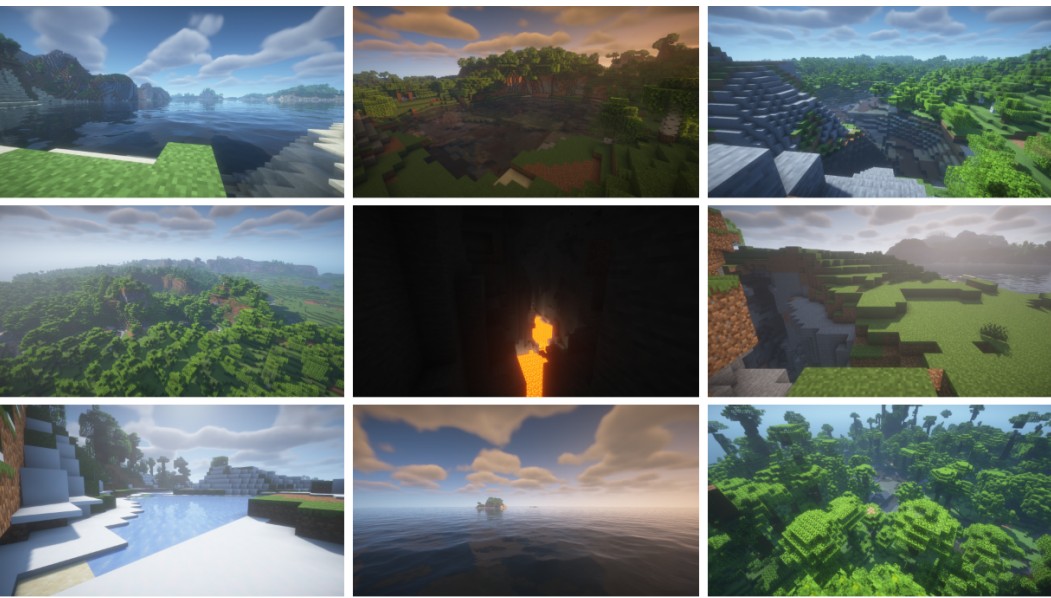

Figure 1: Worldcraft enables the generation of large scale 3D interactive environments. Each image is a screenshot taken from generated world rendered in Minecraft. The world contains over 1 million 3D assets and is a fully interactive game environment. In our context an asset is not a single block but a collection of 4096 blocks termed a chunk.

# 1 INTRODUCTION

Digital media and simulators often require large scale interactive 3D environments comprised of thousands of individual assets. While there has been recent progress in generating these *assets* by combining NeRF-like methods (Mildenhall et al., 2021; Yariv et al., 2021; Poole et al., 2022; Jain et al., 2022), generating the *world* they populate remains an open problem. Current methods for world generation rely on explicitly labeled graphs (Li et al., 2019), autoregressive approaches (Ritchie et al., 2019; Paschalidou et al., 2021) or operate on scenes of a fixed size (Tang et al., 2024) which limit their applicabilty to large scale and variably sized environments. More recent frame-by-frame modeling techniques (Bruce et al., 2024; Valevski et al., 2024) sidestep the issue of scale but fail to capture the persistent, explorable nature of a real world.

In this paper, we take the concept of *world* seriously, drawing a clear distinction between a world and the scenes generated by existing methods. We define a *world* as having three essential properties, largely (and usefully) similar to Minecraft's world model: (i) Object Individuality: Each element possesses a unique identity (analogous to a single block in Minecraft); (ii) Collective Semantics: Multiple objects aggregate to form larger, semantically meaningful structures (comparable to a "region" in Minecraft); (iii) Expandability: The generation process is dynamic, capable of creating new areas as they are explored. In contrast to existing methods, our approach can generate worlds comprising over a million elements on a single, consumer GPU. Thus richly complex worlds rather than mere scenes are achievable both quickly and efficiently. These requirements, combined with our achievements in scale and speed, preclude all existing generative methods (to the best of our knowledge) in the field of 3D environment generation, without significant modifications.

At the core of our technical contribution is casting world generation as a set generation problem – this paradigm shift allows us to create entire worlds at scale and speed, while fulfilling the unique properties in our design of a *world*. Inspired by Ashcroft et al. (2023) generating complex vector drawings with an implicit set space, Worldcraft models the distribution of the top-level parameter of a hierarchy, rather than directly learning the log-likelihood of a permutation-invariant sequence. We then transform a sampled parameter through a series of index functions into a set representing the world. This innovative approach enables us to overcome the limitations of existing methods and achieve our ambitious goals in world generation.

It follows that our key technical innovations all directly tackle the challenges of scale, speed, and our defined world properties:

1. Assumption of Exchangeability: We represent worlds as an exchangeable sequences of latent variables, leveraging the General Representation Theorem (De Finetti, 1929; 1970) to implicitly to represent complex joint distributions through top-level parameters of a hierarchy. This enables us to maintain Object Individuality while simultaneously capturing collective semantics, as the hierarchical structure allows for individual elements to be unique yet form coherent, larger structures and the use of latent variables allows for flexibility in the 3D asset representation.

2. Hybrid Implicit Generation Method: We combine a prescribed probabilistic model (Diggle & Gratton, 1984; Mohamed & Lakshminarayanan, 2016), specifically a Denoising Diffusion Probabilistic Model (Ho et al., 2020), with deterministic subjective mapping functions to efficiently generate large, variable-sized worlds. This hybrid approach is key to achieving both the Scale and Speed requirements, allowing us to generate thousands of elements rapidly on a single GPU.

3. Multi-level Indexing Functions: We introduce a system that allows reconstruction of the world hierarchy at multiple levels, enabling flexible generation and manipulation of world subsets. This directly supports Expandability, allowing for dynamic creation of new areas as they are explored.

These contributions collectively ensure that we can generate vast, coherent worlds that meet our key requirements. Our set-based approach facilitates the creation of large-scale environments with Object Individuality, the exchangeable sequence modeling captures Collective Semantics across different scales, and the multi-level indexing enables the Expandability needed for truly interactive and potentially infinite worlds. Finally it is important to emphasize that Worldcraft is not a tailored solution to generating Minecraft environments but a flexible framework with a range of design choices that enable its application to a wide range of platforms.

## 2 3D SCENES: REPRESENTATION AND GENERATION

Implicit functions represent 3D scenes by capturing their underlying volumetric fields from multi-view images through Neural Radiance Fields (NeRFs) (Mildenhall et al., 2021) and SDFs (Yariv et al., 2021; Zhu et al., 2023). This allows for reconstruction of novel views (Mildenhall et al., 2021) and scene geometry (Yariv et al., 2021), optimized (Zhang et al., 2020a) through voxelisation (Sun et al., 2022) and hashgrids (Müller et al., 2022). Flexibility in implicit representations further allows extension to dynamic scenes through time-specific deformation in 3D space (Cai et al., 2022; Liu et al., 2022a) or by adding a new time dimension altogether (Xian et al., 2021; Shao et al., 2023) for 4D representation. Despite offering high-fidelity novel view generation (Mildenhall et al., 2021), implicit radiance fields often suffer from poor surface and geometry reconstruction (Darmon et al., 2022). NeRF like methods have proven powerful building blocks in larger 3D generative systems (Chan et al., 2021; Schwarz et al., 2020; Gu et al., 2021; Liu et al., 2022b; Poole et al., 2022; Chung et al., 2023). Explicit representations for 3D objects popularly as tangible meshes (Liu et al., 2023) accurately capture surface and geometry (Liu et al., 2021). However, extending meshes for 3D scenes is non-trivial as complexity quickly accumulates from connected objects and scene hierarchies. Recent works suggest using explicit 3D gaussians (3DGS) (Kerbl et al., 2023) as a flexible representation for complex scenes through anisotropic splatting. 3D Gaussian Splatting offers much faster rendering and higher fidelity novel view estimations than corresponding implicit NeRF-based approaches. Surface reconstruction from 3DGS scenes is possible with underlying scene SDFs (Chen et al., 2023; Lyu et al., 2024) and constrained gaussian formation (Guédon & Lepetit, 2024). Both explicit and implicit representation of 3D objects allow for a latent representation of the object.

Scene synthesis is the task of generating realistic 3D scenes of distinct objects often from an existing catalogue of 3D assets. Recent approaches have sought to employ a procedural techniques (Qi et al., 2018; Prakash et al., 2019; Devaranjan et al., 2020; Kar et al., 2019). These require specific pre-defined rule sets which can be expensive and time consuming to create. An alternative approach is to model a scene as a graph(Li et al., 2019; Wang et al., 2019; Zhou et al., 2019; Luo et al., 2020; Purkait et al., 2020; Zhang et al., 2020c;b; Keshavarzi et al., 2020; Di et al., 2020; Gao et al., 2024), explicitly modelling the relationship between objects in the scene. Paschalidou et al. (2021) formulated this problem as a set generation problem and employed a permutation autoregressive approach to generate small scale scenes. Diffusion models have also been employed to this effect (Tang et al., 2024; Yang et al., 2024). What differentiates Worldcraft from existing approaches is the scale of the set generation problem, the use of a hybrid implicit generation method and a hierarchy that instead of modelling explicit relationships between objects uses parameters to represent semantic concepts.

## 3 WORLDCRAFT

We propose modelling 3D interactive world $\mathcal{X}$, as an exchangeable sequence of latent variables that represent 3D assets. As 3D objects may have different represenations but can all be endcoded into a latent space this enables a degree of flexibilty with the method.

**Exchangeability:** Exchangeability is the property of a sequence of random variables that joint probability measure does not change based on the order of observation of the random variables.

$$p(x_1, x_2, ..., x_n) = p(x_{\pi_1}, x_{\pi_2}, ..., x_{\pi_n}) \tag{1}$$

This is shown by equation 1 where for all permutations $\pi$ defined on the set $\{1, 2, ...n\}$ the joint probability measure does not change and that this is true for all subsets. Exchangeability forms the basis of the General Representation Theorem (De Finetti, 1929; 1970).

**General Representation Theorem:** This theorem has several versions (De Finetti, 1929; 1970; Hewitt & Savage, 1955; Diaconis & Freedman, 1984; 1987) but the core concept remains the same that given an exchangeable sequence $\mathcal{X} = \{x_i\}_{i=1}^n$ an integral representation of the joint distribution $p(\mathcal{X})$ can be provided as:

$$p(\mathcal{X}) = \int_F \prod_{i=1}^n F(x_i) dQ(F) \tag{2}$$

In which $F$ is an unknown distribution function and $Q(F) = \lim_{n \to \infty} P_n(\hat{F}_n)$ is defined as the limiting measure on the empirical distribution function $\hat{F}_n$. When indexed by some parameter $\theta \in \Theta$

, $\Theta \subseteq \mathbb{R}^n$ the joint distribution may be written as:

$$p(\mathcal{X}) = \int_\Theta \prod_{i=1}^n p(x_i|\theta)p(\theta)d(\theta) \tag{3}$$

The implications of this are that (Bernardo, 1996); for any subset of an exchangeable sequence of real valued random quantities there must exist some parametric model $p(x|\theta)$ labeled by a parameter $\theta \in \Theta$ and there *exists* a probability distribution for $\theta$ with density $p(\theta)$. As a result $p(\theta)$ implicitly represents $p(\mathcal{X})$. This relationship was utilized by Ashcroft et al. (2023) to generate sets of variable cardinality representing complex vector drawings. A set was decomposed into conditionally independent parametric density functions of the sets individual elements. A generative model was then used to generate latent represents of the set which then conditioned a second model which generated elements of the set.

**Hierarchical Model:** A large exchangeable sequence $\mathcal{X}$ can be divided into subsets such that $\mathcal{X} = \{X_i\}_{i=1}^m$ and $X_m = \{x_i\}_{i=1}^n$. Then joint probability measure can then be defined as:

$$p(\mathcal{X}) = \int_{\Theta^1} \prod_{i=1}^m \prod_{j=1}^n p_i(x_{ij}|\theta_i^1)p(\theta_1^1, ...., \theta_m^1)d\theta_1, ..., d\theta_m^1 \tag{4}$$

Each subset is labelled by a corresponding $\theta_k^1$, the first level of hyper parameters in the hierarchical model, and as such we can treat the original sequence as a sequence of exchangeable parameters, $p(\mathcal{X}) = p(\theta_1^1, ...., \theta_m^1)$. The joint probability measure $p(\mathcal{X})$ can then be defined as:

$$p(\theta_1^1, ...., \theta_m^1) = \int_{\Theta^2} \prod_{i=1}^m p(\theta_i^1|\theta^2)p(\theta^2)d(\theta^2) \tag{5}$$

Where $\theta^2 \in \Theta^2$ is an n-dimensional, second level parameter that labels the distribution of random variables $(\theta_1^1, ...., \theta_m^1)$. This process can be repeated to establish an n-level hierarchy such that the distribution $p(\theta^n)$ implicitly represents $p(\mathcal{X})$.

**Implicit Method:** Instead of directly learning the distribution $p(\mathcal{X})$ we learn the distribution $p(\theta^2)$, the distribution of the top level parameter of the hierarchy and use a series of indexing functions to transform this parameter into $\mathcal{X}$. To do this we first divide the world into subsets of spatially similar elements, we term these subsets regions. Each region is modelled as an indexed family such that there exists a parametric surjective mapping function, $f_\alpha(u_i) \to x_i, u \in U^1$ that index the subsets. We define our index set, $U^1$, as Cartesian product of a known set $I^1$ and the parameter $\theta_k^1$ that represents the subset:

$$U^1 = \{(i, \theta_k) \mid i \in I^1 \text{ and } \theta_\in^1 \Theta^1\} \tag{6}$$

Given a value $\theta_k^1$ and a trained function $f_{\alpha^1}(u_i) \to x_i$, we can reconstruct the subset, however we must first obtain this value $\theta_k^1$. The world can be represented as an exchangeable sequence of the parameters that label the distribution of elements in each region such that $\mathcal{X} = (\theta_1^1, ...., \theta_m^1)$. As such we can treat $\mathcal{X}$ as an indexed family comprised of these labelling parameters and repeat the same process as with the regions. As in the previous step we use a parametric surjective mapping function $f_{\alpha^2}(u_i) \to \theta_i, u \in U^2$ to index with sequence. The index set, $U^2$, is defined as the Cartesian product of the top level parameter $\theta^2$ and a new known set $I^2$.

$$U^2 = \{(i, \theta_m^2) \mid i \in I^2 \text{ and } \theta^2 \in \Theta^2\} \tag{7}$$

As with the previous step we may now reconstruct $\mathcal{X}$ provided we have the initial top level parameter of the hierarchy $\theta^2$. In order to reconstruct the entire sequence $\mathcal{X}$ we need some initial condition $\theta_j^2$ that is an implicit representation of the distribution $p(\mathcal{X})$.

We define a generative model over the top level parameters of the hierarchy $\theta^2 \in \Theta^2$: $p_\phi(\theta^2)$ with trainable parameters $\phi$ where $\bar{\mathcal{D}}$ is a dataset comprised of these top level parameters. We realise $p_\phi(\theta^2)$ with a Denoising Diffusion Probabilistic Model (DDPM) (Ho et al., 2020). Specifically we train a parametric noise estimator $\epsilon_\phi$ on the noisy parameters $\theta_t^2 = \sqrt{\alpha_t}\theta^2 + \sqrt{1 - \alpha_t}\epsilon$ for all

---

**Algorithm 1** Worldcraft generation process

---

Sample: $\theta_k^n \sim p_\alpha(\theta^n)$ from the top level of the hierarchy
**for** n in levels **do**:
    **for** $i \in I^n$, $\theta_k^n \in (\theta_1^n, ...., \theta_m^n)$ **do**
        Reconstruct: $f_{\alpha^n}(\theta_k^n, i) = \theta_u^{n-1}$
    **end for**
**end for**
Reconstruct bottom level of the hierarchy:
**for** $i \in I^1$, $\theta_k^1 \in (\theta_1^1, ...., \theta_m^1)$ **do**
    Reconstruct: $f_\alpha(\theta_k^1, i) = x_u$
**end for**
**for** $x_u \in X_K$ **do**
    Decode: $D_\alpha(x_u) = a_i$
**end for**

---

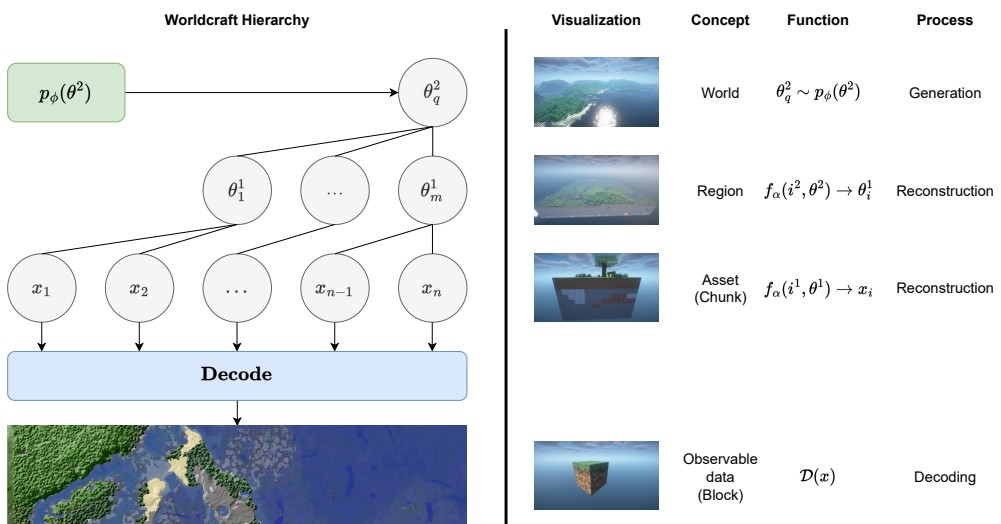

Figure 2: The Worldcraft method has three stages, generation, reconstruction and decoding. In the context of Minecraft a top level parameter is generated that represents the entire world. This parameter labels a model that reconstructs the exchangeable sequence of parameters, each representing a region, that label a second model. Each region is then reconstructed and the resulting latent assets decoded creating the observable world.

$t \in [1, T]$ where $\alpha_t \in [0, 1]$ is a monotonically decreasing diffusion schedule which estimates the noise component $\epsilon$ in:

$$\min_\phi \mathbb{E}_{\theta^2 \in \bar{\mathcal{D}}, \; \epsilon \sim \mathcal{N}(\mathbf{0}, \mathbf{I}), \; t \sim \mathcal{U}(1, T)} \left[ \left|\left| \epsilon_\phi \left( \theta^2{}_t, t \right) - \epsilon \right|\right|_2^2 \right]. \quad (8)$$

We term this a hybrid implicit method as the initial parameter that is transformed into the set $\mathcal{X}$, is sampled from a prescribed probabilistic model $\theta_j^2 \sim p_\phi(\theta^2)$ The generation process for the entire set $\mathcal{X}$ is detailed in Algorithm 1, where $p_{\hat{\alpha}}(\theta^n)$ is a trained generative model, $f_{\hat{\alpha}^n}(i^n, \theta_k)$ is a learned mapping function and $\mathcal{D}_{\hat{\alpha}}(x_i)$ is a decoder taking the latent variable $x_i$ as an input and returning the observable data $a_i$. This algorithm details the full reconstruction of the hierarchy but any subset may be generated through a partial reconstruction. Figure 2 shows the application of the Worldcraft algorithm to Minecraft.

## 4 MINECRAFT

In this section we detail the application of the Worldcraft approach to generating large scale 3D interactive environments. We do so by generating Minecraft worlds. All experiments and results were conducted on a single RTX 4090.

### 4.1 DATA

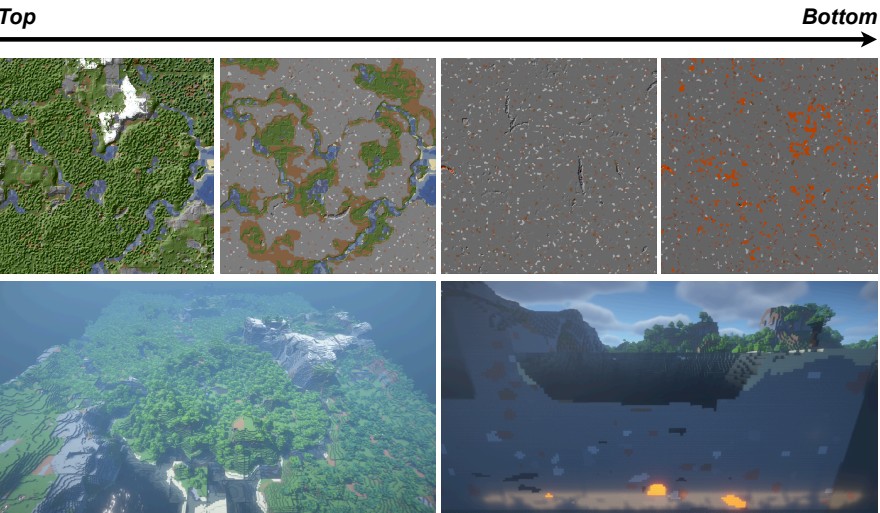

Figure 3: **Top**: Pixel maps of a single generated Minecraft region generated at different elevations. Minecraft worlds are not just surface level features but include complex underground structures from caves to lava lakes. **Bottom**: A top down and side on view of the generated region from inside of the game world show casing the complex 3D structure of a region.

The basic unit of a Minecraft world is a **block**. A block has two components: an ID, which ranges from $0 - 255$ and controls what type of block it is and a state which ranges from $0 - 15$ controlling additional properties of the block. Not all block IDs have 16 block states, invalid combinations of IDs and states result in a block state error, in these cases we use block state 0 for the given ID. A $16 \times 16 \times 16$ group of blocks forms a **chunk**. Chunks always contain 4096 blocks where empty space or air, is represented by an air block. A **region** may have up to $32 \times 32 \times 16$ chunks, this region can be up to a 134 million dimensional object, the make up of a single region is shown in Figure 3. A **world** is comprised of any number of regions arranged in 2D grid. In Minecraft data is stored in files that represent regions and so for the practical reason of easily generating, saving and then loading the data into the game engine we employ the following hierarchy as shown in Figure 2. The entire world is represented by the second level parameter $\theta^2$, a region by $\theta^1$ and each asset by $x_i$ and the blocks by the decoded $x_i$

**Dataset:** The main motivation behind using Minecraft as a test case for the Worldcraft method is the ability to utilize its procedural generation to create a large scale 3D world. We generated two datasets, the first was comprised of 2 million chunks to train the asset (chunk) encoder. The second was a Minecraft world of 1200 regions consisting of 9.8 million chunks. For the full process of how generate and convert the Minecraft world into usable 3D data please refer to Appendix A.1.

### 4.2 ASSET ENCODING

The first step in our method is encode the 3D asset, in this case a chunk, into a latent representation. To do so we employ a 3D convolutional variational autoencoder. We represent each chunk as two channel cube, with the first channel corresponding to the block ID and the second channel the block state. We train the model with a loss function comprised of three components, a weighted KL regularization term, a cross entropy loss for the block-id channel and a cross entropy loss for the block state channel. Our model was trained for 800k steps with a batch size of 128 and constant learning rate of 0.0002.

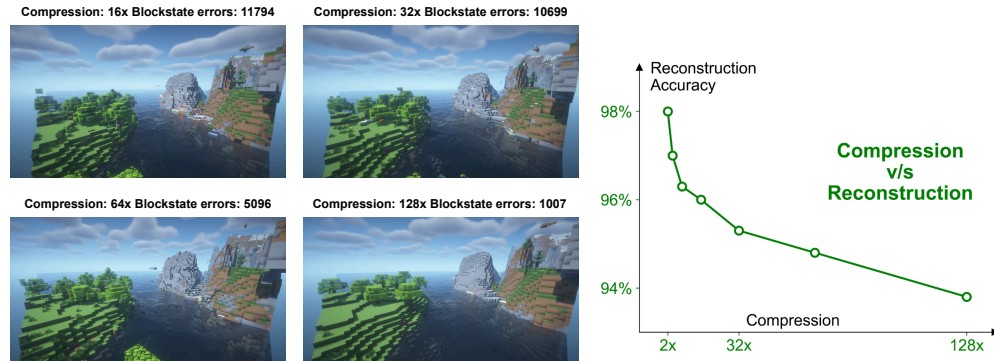

Figure 4: A comparison of the reconstruction accuracy of the chunk encoder at different bottle neck dimensions. **Left**:The same region was passed through autoencoders with different bottleneck size. While assets encoded into higher dimensional spaces have a higher accuracy and include more fine grain details such as foliage, they are prone to more block state errors and erroneous blocks appearing. This can be seen in the top two screenshots where the terrain on the left is more complex and features small bushes. However lava, ice and other erroneous blocks begin appearing.

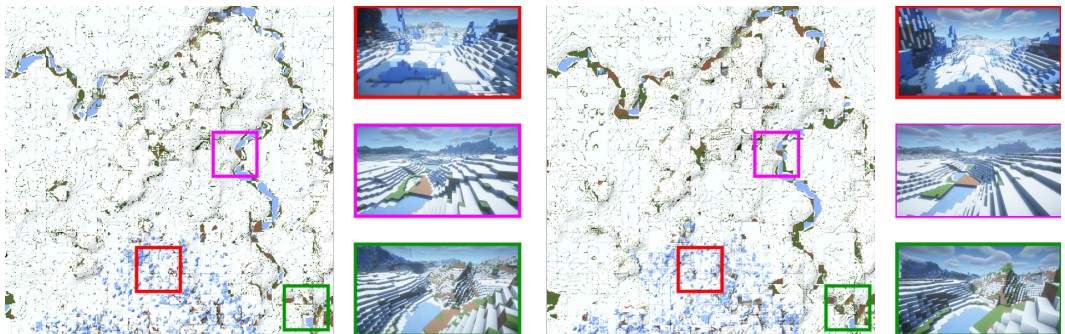

Figure 5: Each pixel map shows the same region reconstructed with different models. The colored squares indicate the area in the region that the corresponding screenshot was taken. The pixel map shows that the terrain from the Large model is more complex and higher in details than the Small model. This can be more easily seen in the screenshots where the Small model omits details such as lava caves, trees etc.

We compared the reconstruction accuracy of chunks at different compression levels as show in Figure 4. While there is a trade off between the reconstruction accuracy of each chunk and the size of the bottle neck in the autoencoder a further consideration is the visual quality. Lower dimension latent representations while providing less fine grain detail were more stable than their high dimensional counter parts. We found a suitable balance between compactness, detail and stability to be at 128. Each region is then represented as an exchangeable sequence $X_k$ of 128 dimension latents.

## 4.3 RECONSTRUCTION

Our goal in reconstruction is that given some parameter $\theta_k \in \Theta$ is to reconstruct the exchangeable sequence that parameter implicitly represents. For a controllable reconstruction process we include the use of an an index set $I$ such that we want to learn some function $f_\alpha(i, \theta_k) \to x_i$ that maps from $i$ and $\theta_k$ onto $x$.

**Indexing**: While a Minecraft region is a $32 \times 16 \times 32$ 3D grid of chunks, above 8 chunks in $y$ it is almost exclusively empty space. As such we define an index set $I = \{(x, y, z) | x \in X, y \in Y \text{ and } z \in Z\}$ resulting in a set with a maximum cardinality of 8192. We scale each index to between zero and one. Each value is then passed through a fourier transform to obtain an n-dimensional positional embedding. In this case we are using the indexing of each element in the sequence to assign it a position in the region.

| Model | Parameters (M) | Time (s) | Block State Errors |
|---|---|---|---|
| Small | 51 | 1.19 | 2635 |
| Medium | 93 | 1.28 | 3115 |
| Large | 126 | 1.35 | 3557 |

Table 1: We compare the reconstruction time for a region given a latent variable $\theta^1$ for the three different models alongside the block state errors. As with the chunk encoding regions with more fine grain details have a larger number of block state errors.

$\theta_\mathbf{k}$: We use a learnable dictionary of embeddings as parameters to represent each region. We thus use a dictionary of 1200, 256 dimension embeddings each corresponding to a unique region.

**Model**: For the reconstruction model we use a multi-layer perceptron. The positional embedding are concatenated with the parameter $\theta_k$ and then input to the model. The model is trained with an MSE loss regularized by a scaled KL-loss. We trained three different region reconstruction models, the properties of which are shown in Table 1. Figure 5 shows the visual differences in reconstruction between then Small and Large model. The Small and Medium models were trained for 3.5k epochs with a batch size of 4096 and a learning rate of 0.001 that we decayed before the end of training. The Large model had a similar scheme but was trained for 5k epochs. Figure 7 shows linear interpolation performed between two parameters with the Large model.

**Latent reconstruction**: We follow a similar reconstruction process for the sequence of parameters with the exception that we use a 2D indexing as a world is comprised of regions assigned to a position in a 2D grid. We created 200 worlds comprised of 144 regions out of the original data set.

**Alternative Indexing**: Minecraft is a dense 3D grid of chunks, for which we know all possible asset positions. This enables us to efficiently explicitly index the set. However in 3D spaces that are continuous this option is less effective as it requires us to either explicitly train the model to assign a latent variable to each possible point in the space or to learn continuous representation. There is however an alternative indexing method that does not require us to know the potential position of an object in advance. If we define our index set in a 1D space and add the position (and any other meta information) of the asset as an output of the model then we are able to not only model continuous 3D spaces but also we do not need to train our model on any empty space.

### 4.4 WORLD GENERATION

The final stage in applying the Worldcraft method is to train a generative model over the parameter space of the latent reconstruction model which then acts as an implicit probability measure over the entire world. To do so we use a Denoising Diffusion Probablistic Model (DDPM) and the same MLP based network architecture as Ashcroft et al. (2023). Once the model is trained we can then perform the full process by sampling from the model, reconstructing the latent sequence, reconstructing the sequence of encoded assets and finally decoding these latent variables into observable data. This process takes 225s for a world comprised of 144 regions. There is an increased amount of time per region as the data needs to saved before rendering. A full generated world can be seen in Figure 6.

### 5 CONCLUSION

We have presented Worldcraft, a novel and flexible framework for generating large scale interactive 3D environments that satisfies our three conditions for a world: Object Individuality, Collective Semantics and Expandability. Instead of directly learning the distribution of a large set of objects we learn the distribution of the top level parameter of the hierarchy and then reconstruct the set through a series of mapping functions. By prioritizing worlds over scenes, this potentially revolutionary set-based approach enables us to generate large scale interactive 3D environments comprised of over a million distinct objects on a single consumer grade GPU, with possible applications across a range of platforms.

A limitation of Worldcraft is that it requires large amounts of training data. Data scarcity is already a challenge in 3D generation tasks but by representing entire worlds of over a million objects as a single sample for generative model learning the distribution of the top level parameters this exacerbates the problem. The method was only tested on Minecraft however we argue with suitable index functions and asset encoding it may be applied to a range of different platforms. In future work directly testing this method on other types of 3D data and addressing the data scarcity will be key.

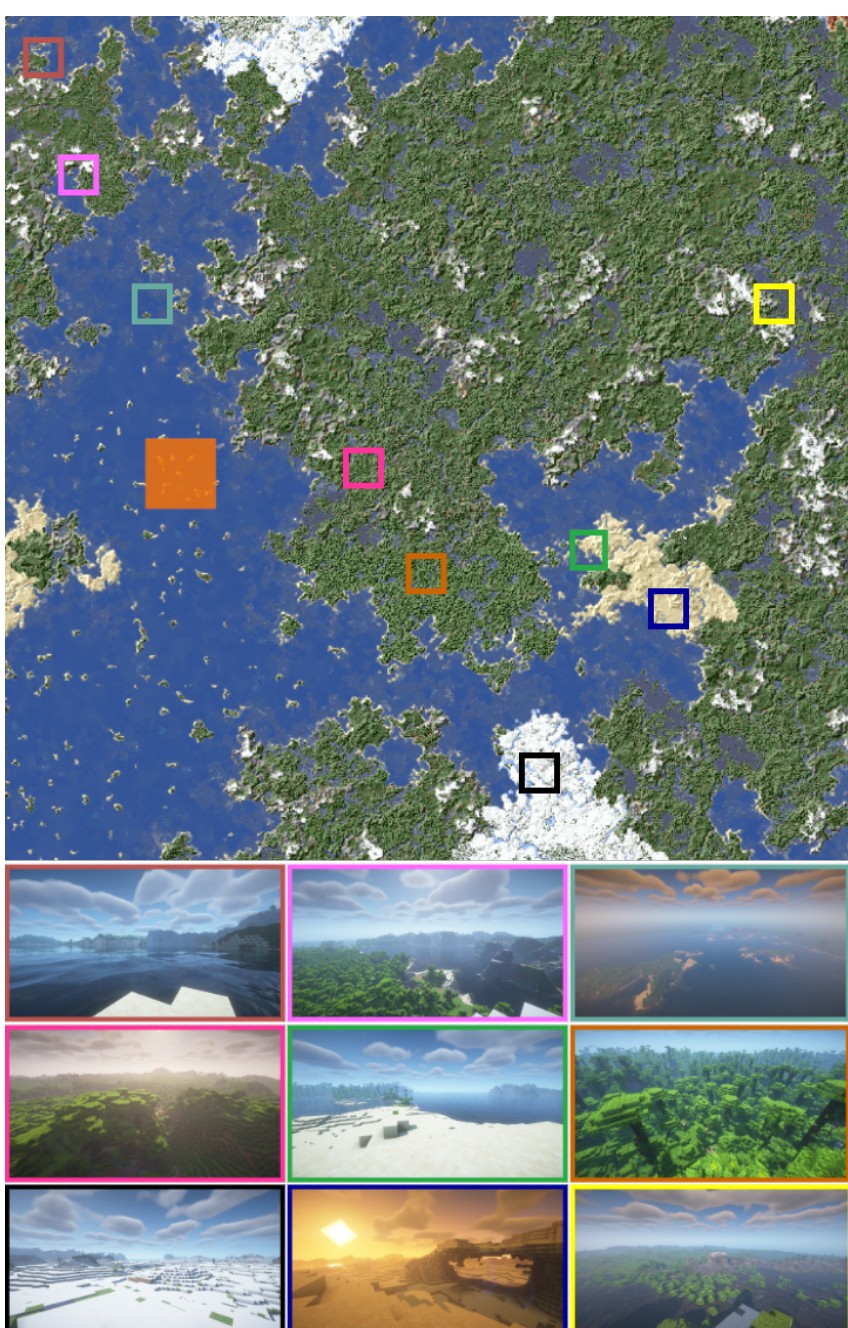

Figure 6: **Top**: We show a scaled down pixel map of a generated world. The orange square indicates a single region as shown in Figure 5. Each small colored square indicates the area in which the screen shorts were taken. **Bottom**: Screenshots taken from withing the generated world. The colored box indicates the area in which they were taken.

**Linear Interpolation**

$\theta_1^1$ $\longrightarrow$ $\theta_2^1$

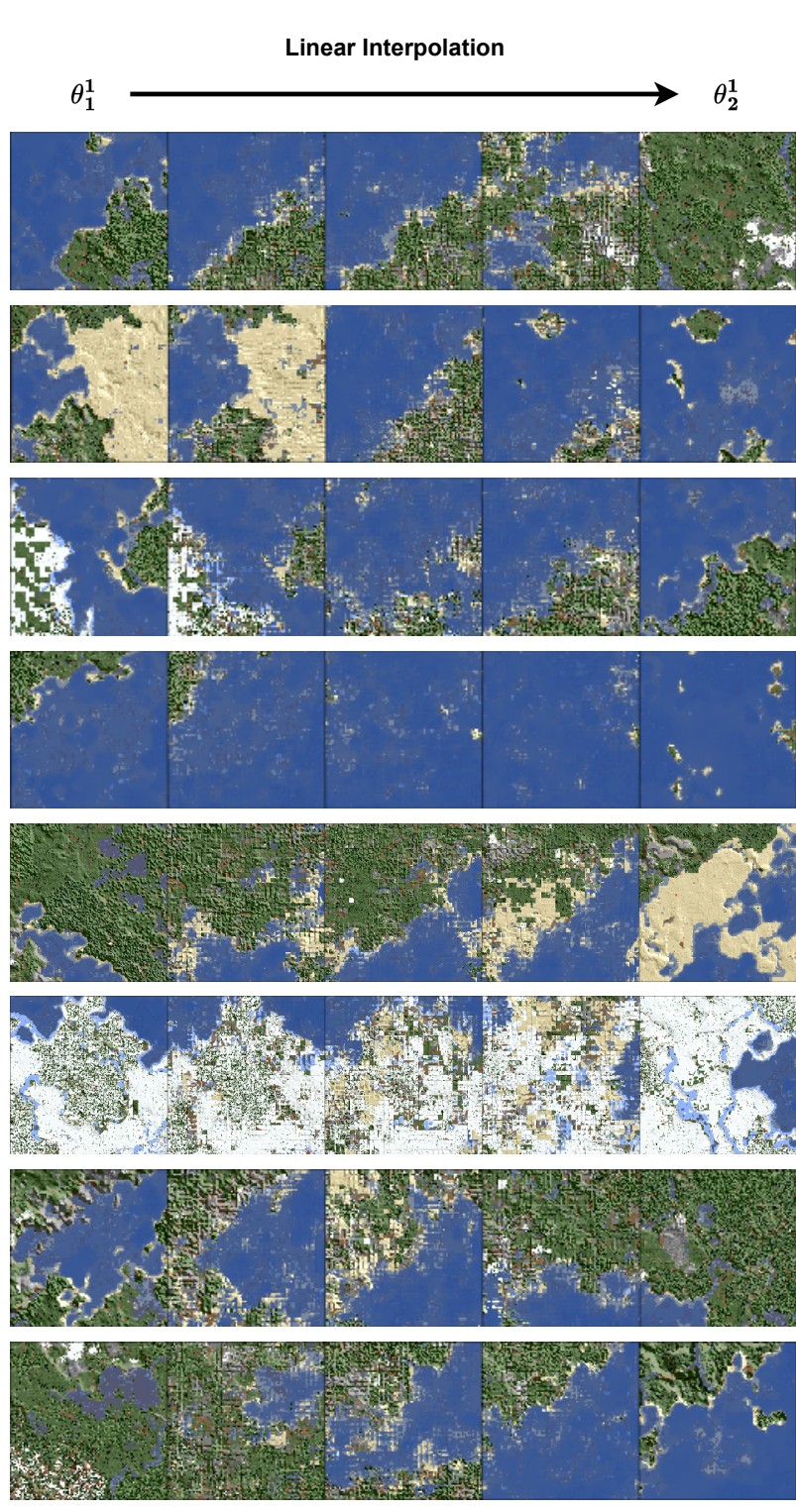

Figure 7: These pixel maps show the surface level features of regions reconstructed for a given $\theta^1$ obtained by linearly interpolating between two samples.

## 6    ETHICS STATEMENT

All training data generated, and rendered results were done in a valid copy of Minecraft. Generative models carry with them several ethical concerns and while our models are trained on and generate Minecraft data the application of our methods to other generative tasks by bad actors may result in disinformation or other damaging content or be used to violate copyright.

## 7    REPODUCIBILTY STATEMENT

We detail in Appendix A.1 the tools used to extract Minecraft data for training the model as well as the method used to generate this training data. Minecraft worlds are by the nature random so there may be slight variations in reproducing the results by generating the data. In addition we detail the training procedures for the models used in this paper either directly or by referring to the scheme used by other authors

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

## A APPENDIX

### A.1 MINECRAFT

On this section we provide specific details about Minecraft and its use in this work. Each world in Minecraft is given a name when created. In the game directory a sub directory with that name will contain a regions sub directory. This directory contains .MCA files. Each .MCA file contains all of the chunks in a region. By geenrating an empty world this sub directory will be empty and we can place our generated data in there to render the data.

**Game version**: We used Minecraft version 1.12.1 to generate our data. The reason for this is that more modern versions of Minecraft swapped from the block state, block ID system for determining blocks and simply assign a block an ID between $0-4095$. We rendered all of our results in Minecraft 1.2.1

**Rendering**: All of the results rendered in this paper were done with BSL shaders on. This is not a requirement.

**Data Generation**: Minecraft generates new terrain within a range of the player in the world. To save the potentially hundreds of hours it would take to exhaustively explore the expansive world we used as a data set we used a world pregenerator mod. This causes the world to generate without the need for manually moving around the world. It took roughly 24 hours to generate the world.

**Data extracting**: To convert the Minecraft data from the .MCA files into usable data we use the anival-parser python library and save the data numpy arrays, where each array contains the entire region.

### A.2 SAMPLES

We provide additional samples from generated Minecraft worlds.

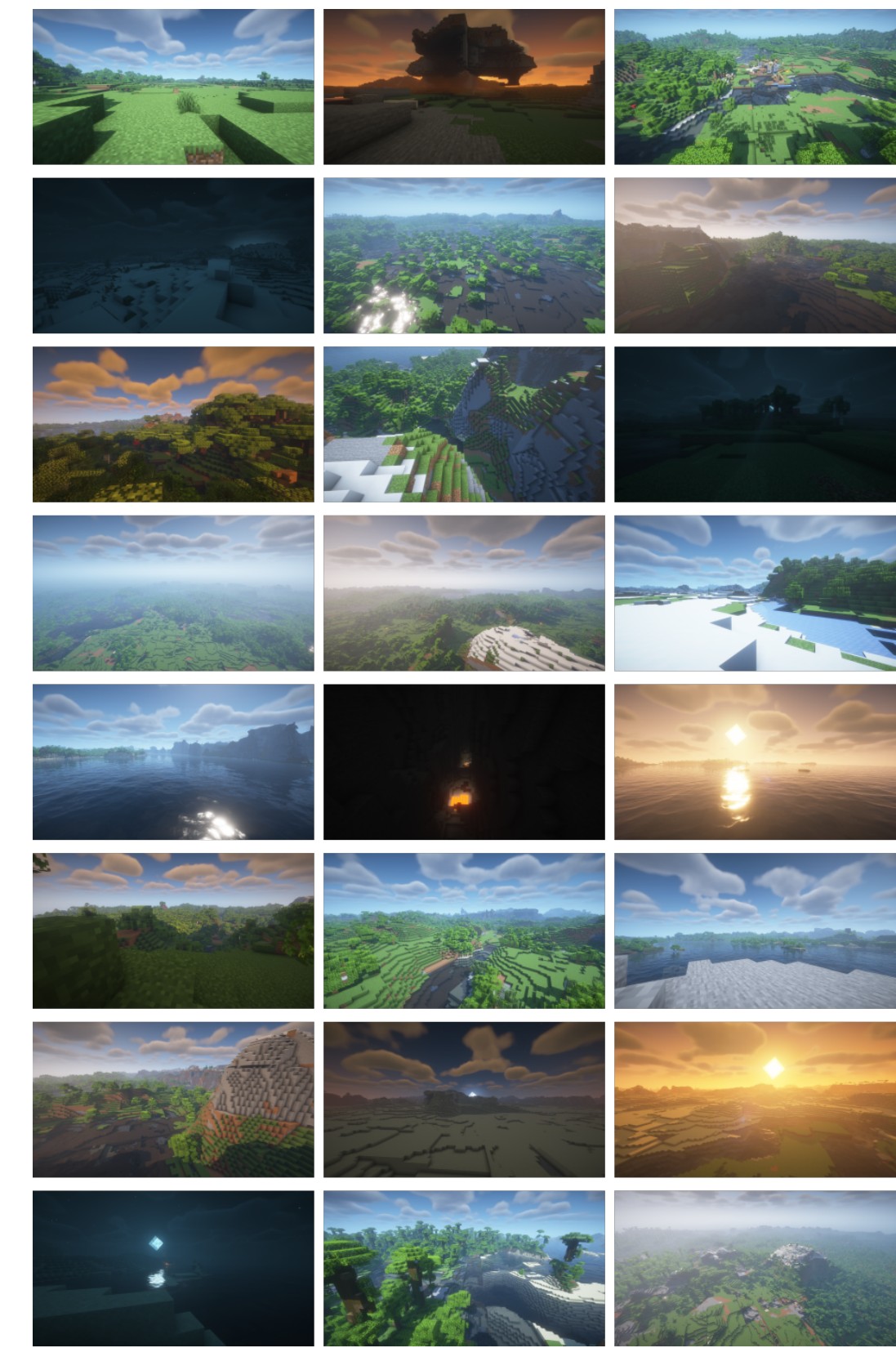

Figure 8: Additional samples taken from a generated world.

