# OpenReview forum: "Worldcraft"
_ICLR.cc/2025/Conference — ICLR 2025 Conference Withdrawn Submission_

### Official Review · Reviewer_byRF · 2024-10-28

**Soundness:** 2
**Presentation:** 2
**Contribution:** 2
**Rating:** 3
**Confidence:** 4

**Summary:**

The paper presents a method for large-scale 3D scene arrangement generation by representing the entire 3D world in a hierarchical tree-based representation, where the leaf nodes are the observable data unit that can be decoded into 3D visual content. But note that, the decoding process is not learned but relies on a predefined renderer (e.g., Minecraft as a game engine). Specifically, the entire 3D world is structured in a coarse-to-fine hierarchy, where the top layer contains high-level semantics and groups of low-level blocks. The paper instantiates its idea on top of Minecraft with a two-layer hierarchy.

**Strengths:**

The paper presents an interesting perspective for large-scale 3D scene layout/arrangement generation through hierarchical designs.

**Weaknesses:**

- The proposed method is limited in both theory and application.
    - Theoretically, the proposed method makes a very strong assumption that the parameter of the decoding process is known prior to the hierarchical modeling of the entire 3D world. This is an assumption that is too strong to be true technically. As for 3D scene generation, the decoding process always indicates the process that decodes the underlying representation to pixels, which is the primary focus for the majority of papers in the field of 3D generation. However, in this paper, the authors assume the decoder is known (at least through the demonstration on top of Minecraft, the decoder is known).
    - Application-wise, the authors use Minecraft as their proof-of-concept for hierarchical world modeling. Yet, the proposed method still requires lots of data for 3D scene generation. However, the data distribution in Minecraft is bounded by its procedural generation process. As a decoder-free method, there is no way for the proposed method to generate 3D scenes beyond the procedural generation. In other words, I do not see the motivation to use procedural generation itself for both quality and efficiency on Minecraft as a special case. Furthermore, whether the proposed method is applicable to other scenarios and platforms is also unknown.
- The evaluation cannot support the paper’s claim.
    - There is no proper evaluation or comparison against other methods or even some simple baselines to better demonstrate the effectiveness of the proposed method. For example, evaluations against procedural generation is needed as well as ablation studies for the number of hierarchies, the effect of errors on world-level pixel maps, etc.
    - As a generative model (as claimed), metrics like FID are recommended to measure.
- The related work is not properly discussed.
    - The authors seem to omit a majority of papers related with 3D scene generation. There are prior works (like GANcraft and SceneDreamer) working on Minecraft-style world translation and generation. Moreover, there are a couple of works (CC3D, GAUDI) focusing on layout-guided scene generation which shares a similar idea with the proposed method in hierarchical modeling.

**Questions:**

Please refer to the Weaknesses section.

---

### Official Review · Reviewer_CF56 · 2024-11-03

**Soundness:** 2
**Presentation:** 2
**Contribution:** 2
**Rating:** 5
**Confidence:** 3

**Summary:**

The paper uses a Denoising Diffusion Probabilistic Model to predict the distribution of a Minecraft map followed by a generative model, two mapping functions and a decoder to generate a complete Minecraft map. The contribution of the paper is a generative model for these maps. Claims are made that the method may also be adapted to other, even continuous, 3D worlds.

**Strengths:**

- The hierarchical nature of the mapping functions allows for n-dimensional level of detail when constructing the worlds. That is adding details to local assets without breaking the continuity of larger regions or the world.
- The 3D Variational Autoencoder used to encode the assets appears to work effectively, even at high compression rates.
- The definition of *world* as it is introduced in the paper separates nicely from "scenes" in 3D reconstruction.

**Weaknesses:**

- The Conclusion argues that the method may be adopted for other, even continuous worlds, but the paper makes little effort to show how this would be done or that it is even possible.
- The use of "unprecedented scale and speed [...]" in the abstract must be specified. Clearly it is not entirely unprecedented as Minecraft itself is already capable of generating such worlds. So what exactly does unprecedented refer to?
- Some of the images are very dark (Fig. 1 Center Image and two Images in Fig. 8).
- Fig. 7 could make use of less examples while giving each example more space to allow for larger images.
- Fig. 6 uses a hard to differentiate color-coding, this hampers accessibility.
- Fig. 6 also contains a large generation artifact in the form of an entirely lava chunk. This remains entirely unaddressed in the paper.
- Fig. 4 shows too small images.
- Fig. 5 uses a primarily white image on a white background. This limits the comparison. The figure also lacks image-to-caption labels.
- Chapter 3 "Worldcraft" mixes statements of previous works (e.g., General Representation Theorem), descriptions of properties (e.g., Exchangeability) and definitions (e.g., their Implicit Method in one large description block).
- The citation style is too verbose.
- The paper fluctuates between using *world* in cursive, as to the own definition of the authors and "world" without cursive. It is however not always clear which is meant when. Perhaps a different term should have been chosen or the paper should follow more carefully the chosen formatting.
- The paper is unscientific in it's choice of words at times:
	Line 64: "In this paper, we take the concept of *world* seriously". What does "taking it seriously" mean?;
	Line 150: "This is shown [...]" when there is just an example/demonstration of Exchangeability and no proof of any sorts;
	Line 213: "We *realise* p_phi(Phi^2) with a Denoising [...]" Generate? Predict?; Realise is an odd choice of word here;
- The title "Worldcraft" is overly vague and all encompassing. It is followed by a general throughline of hubris in the paper, notable examples including:
	Line 64: "In this paper, we take the concept of *world* seriously";
	Line 69,70: "Thus richly complex worlds rather than mere scenes are achievable [...]";
	Line 75: "At the core of our technical contribution [...] this paradigm shift [...]";
	Line 423: "This potentially revolutionary set-based approach [...]".
- Line 303 claims a chunk is a 16x16x16 block. This appears to be incorrect: a chunk extends from the bottom to the top of the world, making it a 16x16x384 block.

**Questions:**

- What are the model architectures for the DDPM, generative Models and the 3D Variational Autoencoder described in the paper?
- Can you prove the claims made in Line 402-406, this seems very relevant to generating continuous scenes?
- How would this method be adapted to worlds that don't already possess a suitable world generation method to create training data -> if a generator is required what benefits does the your model have over just using the generator?
- Minecraft world generation already uses multiple layers (octaves) of differing noise to generate it's world. Is it possible the DDPM merely predicts the combination of these octaves while the hierarchical generative models separate them into their individual noises?
- Why were shaders used in the rendering of the images?

**Details Of Ethics Concerns:**

No ethical concerns.

---

### Official Review · Reviewer_V6Bh · 2024-11-03

**Soundness:** 3
**Presentation:** 2
**Contribution:** 2
**Rating:** 3
**Confidence:** 3

**Summary:**

This paper present Worldcraft, which is a hybrid implicit method for generating vast and interactive 3D worlds at an unprecedented scale and speed. The authors propose to use DDPM as the hybrid generative model to generate different size of worlds with thousands of elements and even up to a million objects in seconds on a single GPU.  The resulting worlds have properties like Object Individuality, Collective Semantics, and Expandability. To achieve this, the authors introduce a hierarchical system and threat the tasks of world generation as a set generation problem and introduces three key technical innovations: Hierarchical and Exchangeable Sequence Modeling, Hybrid Implicit Generation Method, and Multi-level Indexing Functions. The paper demonstrates the idea using Minecraft as a test-bed and present some results using the proposed generative model.

**Strengths:**

1. The pipeline in this submission is technically sound and is clearly written and organized.

2. The authors decompose the task of generating large-scale scenes into multi-scale generation tasks and utilize the DDPM model to generate MineCrafts, which is a cool idea. Additionally, some experimental results prove the efficiency of this method, capable of generating large-scale scenes in about a second.

**Weaknesses:**

1. Overall, the entire paper was difficult for me to understand. The authors claim to have three main contributions, but there is a lack of necessary explanations and experiments to support these claims. Using the DDPM diffusion model to generate MineCraft maps does not support the authors' proposed concept of worldcraft. Additionally, using hierarchical generation models in scene generation tasks is also a quite common strategy.

2. This paper lacks necessary comparative experiments and ablation studies to show the effectiveness and the importance of the proposed method. For comparative experiments, it can directly compare with traditional procedural modeling methods, or compare by generating height maps. For ablation

3. I think the authors have omitted some relevant references on related works regarding 3D scene/world generation like GANCraft, SceneDreamer, InfiniteGen and etc.

[1] Zekun Hao, et. al., Unsupervised 3D Neural Rendering of Minecraft Worlds, ICCV 2021

[2] SceneDreamer, et al., SceneDreamer: Unbounded 3D Scene Generation from 2D Image Collections, TPAMI 2023

[3] Raistrick, Alexander, et al., Infinite Photorealistic Worlds Using Procedural Generation, CVPR 2023

**Questions:**

The authors claim that one of their contributions is the "Assumption of Exchangeability," but aside from a mention in Chapter 3, this term is barely discussed elsewhere. I find it difficult to understand this concept; isn't it just some form of randomness?

---

### Author Response · Authors · 2024-11-18

We appreciate the reviewers' attention but suspect our paper's ambition wasn't fully conveyed. Generating truly interactive worlds requires preserving object individuality - a fundamental departure from purely visual approaches (which the reviewers highlighted for comparison). While unorthodox, this choice was deliberate and, we believe, necessary and timely to steer the community.
## Core Approach
Our method models worlds as interactive object sets because interactivity demands it. Visual fidelity alone, however impressive, cannot enable genuine gameplay. We:
* Implicitly learn distributions over variable-cardinality sets
* Employ Bayesian hierarchies with exchangeability
* Enable deterministic reconstruction
## Why Minecraft Matters
The ability to insert a model's output directly into a game engine allows us to leverage existing tools to provide a high quality result. Our method allows us to use shaders, different graphical settings or any variation the game engine allows. We consider this to be an advantage over purely visual methods.
## On Related Works
We respectfully disagree with comparing our work to visual scene generation methods. The goals are fundamentally different - visual methods focus on appearance, while we focus on function. This isn't a matter of better or worse, but of distinct objectives that serve different purposes. Our method prioritizes interactive functionality, which requires an entirely different technical approach.
## Technical Merit
The exchangeability property and hierarchical structure aren't mathematical flourishes - they're necessary tools for generating worlds where objects maintain their individual identity and function. This enables genuine interaction, which remains the true test of world generation.

---

### Note · Authors · 2024-11-18

I have read and agree with the venue's withdrawal policy on behalf of myself and my co-authors.